# Role of Preoperative Esophagogastroduodenoscopy (EGD) in Bariatric Treatment

**DOI:** 10.3390/jcm10132982

**Published:** 2021-07-03

**Authors:** Regina Sierżantowicz, Jerzy R. Ładny, Krzysztof Kurek, Jolanta Lewko

**Affiliations:** 1Department of Surgical Nursing, Medical University of Bialystok, 15-274 Bialystok, Poland; 2Department of General and Endocrine Surgery, Medical University of Bialystok, 15-276 Bialystok, Poland; ladnyjr@wp.pl; 3Department of Gastroenterology and Internal Medicine, Medical University of Bialystok, 15-276 Bialystok, Poland; krzysztof.kurek@umb.edu.pl; 4Department of Primary Health Care, Medical University of Bialystok, 15-054 Bialystok, Poland; jola.lewko@wp.pl

**Keywords:** preoperative esophagogastroduodenoscopy (EGD), morbid obesity, bariatric surgery

## Abstract

**Background:** The recommendations for routine preoperative esophagogastroduodenoscopy (EGD) in patients qualified for bariatric surgeries are still a matter of debate. The aim of this study was to analyze the pathologies on preoperative EGD in patients qualified for bariatric surgeries. **Materials and Methods:** This study included 222 patients, divided into two groups. The obesity group consisted of patients with obesity (BMI ≥ 40 kg/m^2^), for whom EGD was a routine part of the preparation for laparoscopic sleeve gastrectomy (LSG). The control group of patients with normal body weight (BMI) qualified for EGD because of gastrointestinal ailments. **Results:** Regarding preoperative EGD in patients qualified for bariatric surgeries, we analyzed the prevalence of endoscopic pathologies in various gastrointestinal tract segments. Patients with obesity were shown to present with esophageal pathologies significantly more often than persons in the control group (*n* = 23, 20.91% vs. *n* = 12, 10.91%, *p* = 0.042). The odds ratio of esophageal pathologies in patients with obesity versus the control group equaled 2.15 (95%CI: 1.01–4.59). In turn, the odds ratio of duodenal pathologies in patients from the control group was 3.31 (95%Cl: 1.16–9.47), which means that persons from this group were approximately three times more likely to be diagnosed with those pathologies compared to obese patients. Moreover, patient sex was a significant predictor of duodenal pathologies, with an odds ratio of 4.03 (95%CI: 1.53–10.61). **Conclusions:** Preoperative EGD can identify a broad spectrum of pathologies in obese patients, which suggests a routine examination before bariatric surgery.

## 1. Introduction

Obesity, defined as a body mass index (BMI) ≥30 kg/m^2^, is a health problem that, according to 2014 estimates, affects more than 600 million adults worldwide [1]. The main cause of obesity is the accumulation of adipose tissue in an amount that considerably exceeds the physiological and adaptive needs of the body. In obesity, both the number and the size of adipocytes increase due to persistent positive energy balance, and the synthesis of adipokines promotes the development of related metabolic disorders [2,3].

Treatment of patients with morbid obesity can be challenging and requires an individualized approach. The primary goal of the treatment is to reduce the patient’s body weight by 10% within three to six months. The most commonly used treatment modalities include lifestyle modification, increase in physical activity, diet, and pharmacotherapy. Bariatric surgery is the most effective method to obtain a sustained reduction in body weight [4]. The most frequently performed bariatric procedures are laparoscopic sleeve gastrectomy (LSG) and Roux-en-Y gastric bypass (RYGB) [5].

The indications for routine preoperative esophagogastroduodenoscopy (EGD) in patients qualified for bariatric surgeries are still a matter of debate. None of the American scientific bodies, such as the American Association of Clinical Endocrinologists, the Obesity Society, and the American Society for Metabolic and Bariatric Surgery have specified universal guidelines regarding preoperative EGD; instead, EGD is recommended in selected cases, i.e., in patients with suspected upper gastrointestinal pathologies [6]. According to the European Association for Endoscopic Surgery (EAES), all patients subjected to bariatric surgeries should be examined for potential pathologies of esophageal, gastric, and duodenal mucosa, as well as for hiatal hernia and colonization with *Helicobacter pylori* [7].

Some studies have demonstrated that routine EGD before a bariatric procedure could identify some pathologies, such as hiatal hernia, esophagitis, ulcers, and tumors [8]. Moreover, research has shown that the outcomes of bariatric surgeries in patients colonized with *Helicobacter pylori* are less favorable, as the presence of this microorganism contributes to a longer hospital stay and poses a risk of carcinomatosis within the intact portion of the stomach. Moreover, colonization with *Helicobacter pylori* may be associated with a more frequent occurrence of postoperative symptoms from the large intestine, most often abdominal pain, nausea, and vomiting [9]. However, the results of some studies are inconclusive on this matter, suggesting that colonization with *Helicobacter pylori* is not related to the incidence of early postoperative complications and, hence, preoperative EGD may not be necessary [10]. Furthermore, the results of some studies seem to support selective EGD, solely in patients presenting with gastroesophageal ailments [11]. However, gastrointestinal surgeons have not reached a consensus regarding the cost-effectiveness of EGD; these controversies constitute the rationale for our present study.

It is essential to diagnose and correct comorbidities to obtain good results in bariatric surgery. Non-invasive tests are used for the subjective assessment of a patient’s health and quality of life. Most often, these are questionnaires, the combined use of which is aimed at improving the detection of various disorders, e.g., voice disorders [12].

The aim of this study was to analyze pathologies found in preoperative EGD in patients qualified for bariatric surgeries. Moreover, this study verified whether patients with morbid obesity and normal body weight differ in terms of the frequency of clinical manifestations and pathologies found during preoperative EGD.

## 2. Materials and Methods

### 2.1. Study Population

This study included 222 patients qualified for EGD admitted consecutively to the Diagnostic and Interventional Endoscopy Unit, University Clinical Hospital in Bialystok in 2018–2019. This unit is the largest center for endoscopy and bariatric surgery in Podlaskie Province, conducting the highest number of endoscopic, bariatric, and metabolic surgeries in northeastern Poland.

Before EGD, a history of gastroesophageal ailments was collected from each patient and the medical documentation was analyzed.

Those patients receiving H2 receptor antagonists, proton pump inhibitors (IPP), or acetylsalicylic acid were instructed to withdraw these medications 14 days before the examination. The medical histories and endoscopic findings were recorded and analyzed separately for the two groups:

Group I (Obesity group): 112 patients with obesity (BMI ≥ 40 kg/m^2^), among whom 78 were women (mean age, 43.0 ± 11.4 years) and 34 were men (mean age, 49.56 ± 15.33 years), qualified for EGD as a routine part of the preparation for LSG. The body composition of the study patients was analyzed through dual-energy X-ray absorptiometry (DXA) with Lunar Idxa (GE Healthcare, Chicago, IL, USA). The study included solely those patients qualified for LSD as, in line with the guidelines of the National Institutes of Health [13], this procedure represents the vast majority (85%) of bariatric surgeries performed at our center (1st Clinic of General and Endocrinological Surgery) [13]. Some patients from the obesity group reported mild gastrointestinal ailments, such as postprandial fullness and regurgitation, during the preparation for EGD. Drugs for reducing the acidity (PPI) were taken by 37 (33.6%) patients. Two patients eventually resigned from bariatric treatment and withdrew their consent for the publication of their EGD findings, ultimately resulting in Obesity group consisting of 110 participants.

Group II (Control group): 110 patients with normal body weight (BMI between 18.9 and 24.9 kg/m^2^), among whom 66 were women (mean age, 44.52 ± 16.34 years) and 45 were men (mean age, 45.68 ± 13.0 years), referred for EGD by a gastroenterologist because of various upper gastrointestinal ailments (regurgitation, nausea, vomiting, epigastric pain/burning sensation, heartburn, morning hoarseness, dry cough, postprandial satiety associated with the sense of gastric fullness, and early postprandial fullness). Drugs for reducing the acidity (PPI) were taken by 64 (58.1%) patients.

### 2.2. Exclusion Criteria

Patients were non-eligible for the study if they had a history of prior bariatric surgery or gastrectomy, reported psychoactive substance abuse, suffered from a non-adequately controlled mental disorder, presented with an advanced malignancy, or did not give their consent for participation in the project.

### 2.3. Ethical Considerations

All of the study procedures were designed, conducted, and presented in line with the provisions of the Declaration of Helsinki. The protocol of this study was approved by the Local Bioethics Committee at the Medical University of Bialystok (protocol no. R-I-002/499/2017) and complied with the good clinical practice guidelines. Written informed consent was obtained from the study patients after familiarizing themselves with the objectives and characteristics of all medical procedures.

### 2.4. Esophagogastroduodenoscopy (EGD)

All EGDs were performed by the same endoscopic team (in the case of patients with obesity, including a bariatric surgeon) to provide uniformity of the reporting. The patients were examined after at least 6 h of fasting. Each examination included inspection of the mucosa and collection of biopsy specimens from the prepyloric area of the stomach (2 cm from the pylorus) and gastric fundus, as well as from any identified pathological structures. Moreover, biopsy specimens for the urease test were obtained from the gastric angle area.

Patients identified as colonized with Helicobacter pylori received 14-day eradication therapy with clarithromycin, amoxicillin clavulanate, and a proton pump inhibitor (standard triple therapy). After the therapy, all patients underwent a urea breath test to verify the effectiveness of Helicobacter pylori eradication. No cases of resistance to the triple eradication therapy were observed.

Among the esophageal pathologies, these listed below were assessed based on the endoscopic pictures only: reflux esophagitis (grade A–D), esophageal varices, and hiatal hernia. Barrett’s esophagus was assessed due to histopathology of the specimens taken in accordance with Seattle protocol. Among the gastric pathologies, all types of gastropathies were assessed based on both endoscopic pictures and confirmed further with histopathology of the specimens taken in accordance with Sydney protocol. From gastric polyps and SMT lesions, specimens were taken for further histopathological assessment. Among the duodenal pathologies, ulcers were assessed based on endoscopic appearance only. Duodenal polyps were bioptated for further histopathological assessment.

All gastric and duodenal polyps were very small (<5 mm), removed completely during EGD and found to be hyperplastic in histopathological assessment. All of SMT were referred for further assessment (endoscopic ultrasonography).

No complications of EGD, whether related to the endoscopic examination or associated with anesthesia (whenever administered), were recorded.

## 3. Statistical Analysis

The normal distribution of quantitative variables was verified with the Shapiro–Wilk test. As the study variables were not distributed normally, the Mann–Whitney xi-test was used for between-group comparisons. Statistical significance of relationships between qualitative variables was verified with Pearson’s chi-squared test. Additionally, odds ratios (ORs) were calculated with their 95% confidence intervals (95%CIs). The significance level was set at 0.95, the results were considered statistically significant at *p* < 0.05.

## 4. Results

A total of 220 patients were included in this study and provided their consent to analyze their results. In the obesity group, women (*p* = 0.048) and men aged 49.6 ± 15.3 (*p* = 0.031) were statistically more common. There were differences between the obesity group and the control group in terms of BMI and WHR (*p* = 0.001). Men in the control group smoked significantly more often. The following diseases were significantly more frequent in the group of obese patients: type 2 diabetes, hypertension, and dyslipidemia. The differences between the groups were statistically significant, the data is presented in Table 1.

In 15/220 patients, the result of the EGD corresponded to reflux esophagitis by the Los Angeles system [14], with four (3.63%), three (2.72%), and one (0.90%) patient(s) with obesity presenting with grade A, B, and D pathologies, respectively. None of the patients were diagnosed with grade C esophagitis. Esophageal varices, small according to the World Gastroenterology Organization (WGO) classification [15], were found in one (0.90%) patient with obesity. Hiatal hernia was more frequent in the obesity group (*n* = 6, 5.41%). Two patients, one (0.90%) from the obesity group and another from the control group, presented with Barrett’s esophagus (Table 2).

EGD demonstrated gastritis in 176/220 patients, 88 (80%) from either group. Patients with obesity were diagnosed with erythematous gastropathy (*n* = 38, 34.23%), erythematous nodular gastropathy (*n* = 16, 14.68%), erythematous erosive gastropathy (*n* = 2, 1.80%), erosive gastropathy (*n* = 11, 9.91%), and atrophic gastropathy (*n* = 6, 5.41%). All of these types of gastropathy were also found in the control group, but with slightly (non-significantly) different frequencies. A total of 36/220 patients, 18 (16.36%) from either group, tested positively for *Helicobacter pylori*. All *Helicobacter pylori*-positive patients received the same standard eradication therapy. In 11/220 patients, EGD revealed gastric polyps and in 3/220 patients, two (1.81%) obese patients and one (0.90%) patient from the control group, presented submucosal gastric tumors. The frequencies of gastric pathologies found using EGD in patients from both groups are shown in Table 3.

In 189/220 patients, EGD did not show any duodenal pathologies. Patients with obesity presented with duodenogastric reflux (*n* = 2, 1.83%) and duodenal ulcers (*n* = 3, 2.72%), and one patient from the control group (0.90%) was diagnosed with a duodenal polyp. The study groups did not differ significantly in terms of the frequency of specific duodenal pathologies found using EGD (Table 4).

Patients with normal body weight and obesity, and women and men were compared in terms of the prevalence of gastrointestinal pathologies found using EGD. To find a justification for preoperative EGD in patients qualified for bariatric surgeries, we analyzed the prevalence of endoscopic pathologies in various gastrointestinal tract segments. Patients with obesity were shown to present with esophageal pathologies significantly more often than persons with normal body weight (*n* = 23, 20.91% vs. *n* = 12, 10.91%, *p* = 0.042). In turn, duodenal pathologies were significantly more frequent in control group patients than in those with obesity (*n* = 15, 13.64% vs. *n* = 5, 4.55%; *p* < 0.019) and were found significantly more often in men than in women (*n* = 13, 17.11% vs. *n* = 7, 4.86; *p* < 0.002) (Table 5).

The odds ratio of esophageal pathologies in patients with obesity versus those from the control group equaled 2.15 (95%CI: 1.01–4.59), which implies that obese persons were twice as likely to present with esophageal diseases using EGD. In turn, the odds ratio of duodenal pathologies in patients from the control group was 3.31 (95%Cl: 1.16–9.47), which means that persons from this group were approximately three times more likely to be diagnosed with those pathologies compared to obese patients. Moreover, patient sex was a significant predictor of duodenal pathologies, with an odds ratio of 4.03 (95%CI: 1.53–10.61), corresponding to a four times more frequent occurrence in men than in women (Table 6).

## 5. Discussion

Currently, bariatric surgeries are the most effective treatment option in morbid obesity [4]. A review of multiple reports published over more than 10 years/with a greater than 10-year observation demonstrated that the percentage of excess weight loss (%EWL) could vary from 56.7% after gastric bypass (RYGB) to 45.9% after laparoscopic adjustable gastric banding (LAGB) and 58.3% after laparoscopic sleeve gastrectomy (SG) [16]. However, bariatric procedures also pose a considerable risk of complications [17,18,19], such as anastomotic leakage [20,21], bleeding [22,23], marginal ulcer [23,24], and anastomotic stenosis [24]. In the present study, 50% (*n* = 110) of the patients had a preoperative BMI ≥ 43.5 ± 4.3 m^2^, corresponding to severe obesity, in the case of which surgical treatment may produce desirable outcomes. Most of the study patients were women: The control group with normal BMI included 66 women with a mean age of 44.5 ± 16.3 years, while the group with obesity included 78 women with a mean age of 43.0 ± 11.4 years; such a distribution of sex among patients qualified for bariatric procedures is consistent with the results of previous studies [8,10,14]. Regional differences in lipolysis between different fat deposits are more pronounced in obese people. In women, the subcutaneous tissue is dominant and usually located in the buttock-femoral area rather than the abdominal area, as in men. Males have larger adipocytes in this deposit, which may be one of the main causes of the gender differences seen in obesity. The obesity group had a higher female proportion (71% vs. 60%), which may be a significant influence on some of the reported data. It should be noted that abdominal obesity and the breakdown of adipose tissue may contribute to the predominance of risk for malignancies, Barrett’s esophagus or reflux, because it is associated with an increased risk of disease, regardless of body mass index (BMI) [25].

An advantage of EGD is the possibility of visualizing esophageal, gastric, and duodenal mucosa, and with the evaluation for *Helicobacter pylori* infection, inflammation, and potential neoplastic lesions. In addition, the complication rate after EGD is approximately 1% [26,27,28]. Examination can provide additional information that warrants qualification for bariatric surgery, selection of its type, and necessity of concomitant surgical procedures (e.g., hiatal hernia repair) and preoperative treatment (e.g., eradication of *Helicobacter pylori*). This makes EGD clinically meaningful as, according to the literature, the prevalence of upper gastrointestinal pathologies in asymptomatic patients is high [29,30]. Our findings confirm that patients with obesity may present with various abnormalities that can be visualized using EGD, such as esophageal (*n* = 23, 20.91%), gastric (*n* = 95, 86.36%), duodenal (*n* = 5, 4.55%), and intestinal pathologies (*n* = 4, 3.64%), as well as *Helicobacter pylori* colonization (*n* = 18, 16.36%). Only a few previous studies pulled into question a correlation between clinical manifestations and EGD findings; in the largest of those studies, a retrospective analysis of the pre- and postoperative role of EGD in 3219 patients with obesity, Abd Ellatif et al. [31] found endoscopic abnormalities in 6% of the cases, which was markedly lower than the 31–89.7% prevalence reported by other authors [6,29,30,32].

Although the frequency of *Helicobacter pylori* colonization has been decreasing worldwide in recent decades, in line with improvements in socioeconomic conditions, the prevalence of this microorganism in the general population may vary. While in Northern Europe and North America, only approximately one-third of adults are colonized with *Helicobacter pylori*, the colonization rates in Southern and Eastern Europe, South America, and Asia frequently exceed 50% [33]. Moreover, the *Helicobacter pylori* colonization rates in patients qualified for bariatric treatment vary considerably across studies, from 23% to 70%; importantly, these values are markedly higher than in our present study [32]. It is estimated that 89% of gastric cancers are associated with chronic colonization with *Helicobacter pylori*; therefore, we considered colonization with this microorganism as a separate entity among gastric pathologies. The proportions of *Helicobacter pylori*-colonized patients from both groups were the same (*n* = 18, 16.36%)—lower than reported previously.

All patients with a positive test for *Helicobacter pylori* underwent effective eradication therapy in accordance with the applicable standards and on the recommendation of the attending physician. However, in most European countries, therapy based on the use of a single capsule of bismuth is indicated and used [34].

In this study, patients from both groups presented with endoscopic pathologies of the esophagus, such as reflux esophagitis of various degrees, esophageal varices, hiatal hernia, and Barrett’s esophagus. The study groups did not differ significantly in the prevalence of the specific esophageal pathologies; however, when the results were analyzed cumulatively, rather than according to the pathology type, the between-group difference turned out to be statistically significant (*p* < 0.042), with the odds of endoscopic esophageal abnormalities in patients with obesity being approximately twice as high as in the normal-weight persons (OR = 2.15, 95%CI: 1.01–4.59, *p* < 0.045). According to DuPree et al. [35], LSG may induce GERD in some asymptomatic patients. During the International LSG Consensus Conference, 52.5% of general surgeons and 23.3% of bariatric surgeons considered GERD a contraindication to LSG [27]. Similarly, the diagnosis of Barrett’s esophagus may influence the choice of bariatric procedure, with RYGB as the preferred option.

The most common endoscopic pathology documented in this study was gastritis, found in 88 (80%) patients from either group; this observation is consistent with the results of previous studies [36]. Gastric polyps are often benign lesions, but may also be precursors for precancerous conditions and gastric cancer. Gastric polyps were found in five (4.54%) patients with obesity, and two (1.81%) persons from this group presented with submucosal gastric tumors (SMT). The prevalence of these pathologies did not differ significantly between the group with obesity and normal body weight. However, according to some authors [36], gastritis found during preoperative EGD may correlate with postoperative anastomotic ulceration. This implies that preoperative diagnosis and treatment of gastritis might contribute to better outcomes of bariatric treatment [37,38]. Among duodenal pathologies, the study patients were diagnosed with duodenogastric reflux and duodenal ulcers. We did not assess the direct numbers of duodenal refluxes during EGD. However, we diagnosed the presence of duodenal refluxes based on the findings in the stomach (e.g., presence of bile in the stomach, incrustation of gastric mucosa with biliary precipitates). Such changes indirectly indicate the presence of duodenal reflux. The overall prevalence of duodenal pathologies was significantly higher in men than in women (*p* < 0.002), with approximately four times higher odds (OR = 4.03, 95%CI: 1.53–10.61, *p* < 0.004) in patients from the control group than in those with obesity (*p* < 0.019).

A strength of this study is the possibility to compare endoscopic findings in obese patients without significant clinical symptoms with the control group of 110 symptomatic patients with normal BMI. The fact that the study included patients from a single center may be a factor limiting the generalizability of the results and, hence, our findings need to be verified in larger, multicenter research. The limitation of the study was the lack of a recommendation to perform a non-invasive test with a negative *Helicobacter pylori* biopsy result. The obesity group had a higher female proportion, which may be significant influence on some of the reported data.

## 6. Conclusions

Preoperative esophagogastroduodenoscopy (EGD) can identify a broad spectrum of pathologies in patients with obesity. Endoscopic diagnosis may influence further therapeutic decisions, including the selection of the most appropriate bariatric procedure. Thus, routine upper gastrointestinal endoscopy before a bariatric procedure may be recommended in future by relevant scientific bodies.

## Figures and Tables

**Table 1 jcm-10-02982-t001:** Demographic data in the group of patients with a normal BMI and the group of obese patients.

	Obesity Group F/M	(*n* = 110)	Control Group F/M	(*n* = 110)	*p*-Value * F/M	General F/M	General (*n* = 220)
Sex (female/male) *n/%*	78 (70.9%)/ 32 (29.1%)	110 (50%)	66 (60%)/ 44 (40%)	110 (50%)	0.048	144 (65.5%)/ 76 (34.5%)	220 (100%)
*Median age (years)* *± SD*	43.0 ± 11.4/ 49.6 ± 15.3	47.3 ± 12.0	44.5 ± 16.3/ 45.7 ± 13.0	45.1 ± 14.7	0.214/0.031	43.6 ± 14.2/ 47.6 ± 14.2	47.0 ± 16.0
*Median BMI (kg/m^2^)* *± SD*	44.1 ± 4.3/42.9 ± 4.3	43.5 ± 4.3	22.7 ± 2.8/ 23.8 ± 2.8	23.3 ± 2.8	0.001/0.001	-	-
*Median WHR* *± SD*	0.95 ± 0.07/ 1.02 ± 0.06	0.96 ± 0.08	0.80 ± 0.08/ 0.87 ± 0.06	0.83 ± 0.11	0.001/0.001	-	-
*%FAT (DXA)*	50.2 ± 3.9/ 45.8 ± 5.5	48.0 ± 4.7	Not tested	-	-	-	-
*Place of residence*							
*Village, n/%*	15 (19.2%)/ 10 (31.3%)	25 (22.7%)	8 (12.1%)/ 6 (13.6%)	14 (12.7%)	0.500/0.001	23 (10.5%)/ 16 (7.3%)	39 (17.7%)
*City with <10,000 inhabitants, n/%*	29 (37.2%)/ 9 (28.1%)	38 (34.5%)	15 (22.7%)/ 9 (20.4%)	24 (21.8%)		44 (20%)/ 18 (8.1%)	62 (28.2%)
*City with >10,000 inhabitants, n/%*	34 (43.6%)/13 (40.6%)	47 (42.7%)	43 (65.2%)/ 29 (65.9%)	72 (65.5%)		77 (35%)/42 (19.1%)	119 (54.1%)
*Education*							
*Secondary, n/%*	7 (8.9%)/5 (15.6%)	12 (10.9%)	3 (4.5%)/ 5 (11.3%)	8 (7.3%)	0.530/0.084	10 (4.5%)/10 (4.5%)	20 (9.1%)
*Vocational, n/%*	5 (6.4%)/8 (25%)	13 (11.9%)	4 (6.1%)/8 (18.2%)	12 (10.9%)		9 (4.1%)/16 (7.3%)	25 (11.4%)
*Bachelor’s, n/%*	37 (47.4%)/8 (25%)	45 (40.7%)	39 (59.1%)/ 13 (29.5%)	52 (47.3%)		76 (34.5%)/21 (9.5%)	97 (44.1%)
*Higher, n/%*	29 (37.2%)/11 (34.4%)	40 (36.5%)	20 (30.3%)/ 18 (40.1%)	38 (34.5%)		49 (22.3%)/29 (13.2%)	78 (35.4%)
*Source of income*							
*Physical work*	12 (15.4%)/20 (62.5%)	32 (29.1%)	14 (21.2%)/20 (45.4%)	34 (30.9%)	0.034/0.082	26 (18.1%)/40 (52.7%)	66 (30.1%)
*Mental work*	17 (21.8%)/8 (25%)	25 (22.7%)	21 (31.8%)/ 17 (38.6%)	38 (34.5%)		38 (26.4%)/25 (32.9%)	63 (28.6%)
*Pupil/student*	12 (15.4%)/2 (6.3%)	14 (12.7%)	12 (18.2%)/4 (9.1%)	16 (14.5%)		24 (16.7%)/6 (7.8%)	30 (13.6%)
*Pension/retired, n/%*	37 (47.4%)/2 (6.3%)	39 (35.5%)	19 (28.8%)/3 (6.8%)	22 (20.1%)		56 (38.8%)/5 (6.6%)	61 (27.7%)
*Smoking*							
*Smoking, n/%*	9 (11.5%)/10 (31.3%)	19 (17.3%)	19 (28.8%)/16 (36.3%)	35 (31.8%)	0.623/0.001	28 (19.4%)/26 (34.2%)	54 (24.5%)
*Non-smoking, n/%*	69 (88.5%)/22 (68.7%)	91 (82.7%)	47 (71.2%)/28 (65.9%)	75 (68.2%)		116 (80.6%)/50 (65.8%)	166 (75.5%)
*Comorbidities*							
*Dysglycemia diagnosis: Non-diabetic, n/%*	12 (15.4%)/ 8 (25%)	20 (18.2%)	8 (12.1%)/4 (9.1%)	12 (10.9%)	0.307/0.556	20 (13.8%)/ 12 (15.7%)	32 (14.5%)
*Diabetes mellitus type 2, n/%*	17 (21.8%)/ 11 (34.4%)	28 (25.5%)	4 (6.1%)/2 (4.5%)	6 (5.5%)	0.037/0.001	21 (14.6%)/ 23 (30.1%)	44 (20%)
*Arterial hypertension, n/%*	19 (24.4%)/ 18 (56.3%)	37 (33.6%)	10 (15,2%)/ 12 (27.3%)	22 (20%)	0.029/0.001	29 (20.1%)/30 (39.5%)	59 (26.8%)
*Obstructive sleep apnea, n/%*	5 (6.4%)/ 9 (28.1%)	14 (12.7%)	1 (1.5%)/3 (6.8%)	4 (3.6%)	0.720/0.007	6 (4.2%)/12 (15.8%)	18 (8.2%)
*Fatty liver (n/%)*	9(11.5%)/ 1(3.1%)	10 (9.1%)	3(4.5%)/ 3(6.8%)	6 (5.5%)	0.247/-	12(8.3%) 4(5.2%)	16 (7.3)
*Dyslipidemia (n/%)*	22 (28.2%)/ 10 (31.3%)	32 (29.1%)	6(9.1%)/ 4(9.1%)	10 (9.1%)	0.005/0.024	28 (19.4%) 14 (18.4%)	42 (19.1%)

BMI, body mass index; WHR, waist–to-hip ratio; FAT% (DXA), percentage of body fat as measured by a dual-energy X-ray absorptiometer; F/M-female/male, Data are presented *n*/%—number/percent, Median ± SD-Median ± Standard deviation; * Pearson’s chi-squared test.

**Table 2 jcm-10-02982-t002:** Esophageal pathologies found via EGD in the study groups.

EGD Finding	Obesity Group	Control Group	*p*-Value *
Reflux esophagitis	6 (5.45%)	9 (8.18%)	0.086
Grade A	4 (3.63%)	5 (4.54%)	0.337
Grade B	3 (2.72%)	2 (1.81%)	0.093
Grade C	0 (0.00%)	0 (0.00%)	ns
Grade D	1 (0.90%)	2 (1.81%)	0.304
Esophageal varices	1 (0.90%)	0 (0.00%)	0.320
Hiatal hernia	7 (6.42%)	2 (1.81%)	0.083
Barrett’s esophagus	1 (0.90%)	1 (0.90%)	0.311

The esophageal pathologies listed below were assessed based on endoscopic pictures only: reflux esophagitis (grade A–D), esophageal varices, and hiatal hernia. Barrett’s esophagus was assessed by histopathology of the specimens taken in accordance with Seattle protocol. * Pearson’s chi-squared test.

**Table 3 jcm-10-02982-t003:** Gastric pathologies found using EGD in the study groups.

EGD Finding	Obesity Group	Control Group	*p*-Value *
Hp+	18 (16.36%)	18 (16.36%)	1.000
Chronic gastritis	88 (80%)	88 (80%)	1.000
Erythematous gastropathy	38 (34.23%)	45 (41.28%)	0.280
Erythematous nodular gastropathy	16 (14.68%)	11 (9.91%)	0.281
Erythematous erosive gastropathy	2 (1.80%)	7 (6.42%)	0.083
Erosive gastropathy	11 (9.91%)	16 (14.68%)	0.281
Atrophic gastropathy	6 (5.41%)	8 (7.34%)	0.761
Gastric polyp	5 (4.54%)	6 (5.45%)	0.536
Submucosal gastric tumor (SMT)	2 (1.81%)	1 (0.90%)	0.304

The gastric pathologies of all types of gastropathies were assessed based on both endoscopic pictures and confirmed further with histopathology of the specimens taken in accordance with Sydney protocol. From gastric polyps and SMT lesions, specimens were taken to further histopathological assessment. * Pearson’s chi-squared test.

**Table 4 jcm-10-02982-t004:** Duodenal pathologies found using EGD in the study groups.

EGD Finding	Obesity Group	Control Group	*p*-Value *
Duodenogastric reflux	2 (1.83%)	7 (6.42%)	0.083
Duodenal ulcer	3 (2.72%)	7 (6.42%)	0.151
Duodenal polyp	0 (0.00%)	1 (0.90%)	0.311

The duodenal pathologies ulcers were assessed based on endoscopic appearance only. Duodenal polyps were bioptated for further histopathological assessment. * Pearson’s chi-squared test.

**Table 5 jcm-10-02982-t005:** Prevalence of gastrointestinal pathologies in obese group of patients and control group of patients and in women and men.

	**Obesity Group**	**Control Group**	***p*-Value ***
Hp+	18 (16.36%)	18 (16.36%)	1.000
Hp−	92 (83.64%)	92 (83.64%)	
	**F**	**M**	
Hp+	20 (13.89%)	16 (21.05%)	0.172
Hp−	124 (86.11%)	60 (78.95%)	
	**Obesity group**	**Control group**	
Esophageal pathologies +	23 (20.91%)	12 (10.91%)	0.042
Esophageal pathologies −	87 (79.09%)	98 (89.09%)	
	**F**	**M**	
Esophageal pathologies +	26 (18.05%)	9 (11.84%)	0.230
Esophageal pathologies −	118 (81.94%)	67 (88.16%)	
	**Obesity group**	**Control group**	
Gastric pathologies +	95 (86.36%)	95 (86.36%)	1.000
Gastric pathologies −	15 (13.64%)	15 (13.64%)	
	**F**	**M**	
Gastric pathologies +	125 (86.81%)	65 (85.53%)	0.793
Gastric pathologies −	19 (13.19%)	11 (14.47%)	
	**Obesity group**	**Control group**	
Duodenal pathologies +	5 (4.55%)	15 (13.64%)	0.019
Duodenal pathologies −	105 (95.45%)	95 (86.36%)	
	**F**	**M**	
Duodenal pathologies +	7 (4.86%)	13 (17.11%)	0.002
Duodenal pathologies −	137 (95.14%)	63 (82.89%)	

F-female, M-male; * Pearson’s chi-squared test.

**Table 6 jcm-10-02982-t006:** Predictors of gastrointestinal pathologies using EGD.

Predictor/Outcome	Odds Ratio (OR)	Lower Bound (95%CI)	Upper Bound (95%CI)	*p*-Value *
Obesity/esophageal pathologies	2.159	1.014	4.594	0.045
Obesity/duodenal pathologies	3.315	1.160	9.470	0.025
Sex/duodenal pathologies	4.038	1.536	10.611	0.004

OR, odds ratios; 95%CI, 95% confidence interval; * Pearson’s chi-squared test.

## Data Availability

The data presented in this study are available on request from the corresponding author. The data are not publicly available due to ethical and privacy restrictions.

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
