# Peer review of "Role of Preoperative Esophagogastroduodenoscopy (EGD) in Bariatric Treatment"

_jcm, 2021, doi:10.3390/jcm10132982_

Round 1
Reviewer 1 Report
Good work well written. i report my suggests:
- the authors should review english
- The authors have not complications using EGD but to be complete in the description they should report in the abstrac/discussiont the percentage of complications of EGD in the literature
- the authors with EGD detecte comorbidities to have good results with bariatric surgery. To be complete, the authors should mention the possibility using some test not invasive (e.g. test, questionaire). ABout this i suggest you to mention in your bibliography this work:
Galletti B, Sireci F, Mollica R, Iacona E, Freni F, Martines F, Scherdel EP, Bruno R, Longo P, Galletti. Vocal Tract Discomfort Scale (VTDS) and Voice Symptom Scale (VoiSS) in the Early Identification of Italian Teachers with Voice Disorders. Int Arch Otorhinolaryngol. 2020;24(3):e323-e329
Author Response
Responses for reviewer
Reviewer 1
- The authors should review English
Yes, we use MDPI English editing Serwis
- The authors have not complications using EGD but to be complete in the description they should report in the abstrac/discussiont the percentage of complications of EGD in the literaturę
…the complication rate after EGD is approximately 1% [25,26,27].
- The authors with EGD detecte comorbidities to have good results with bariatric surgery. To be complete, the authors should mention the possibility using some test not invasive (e.g. test, questionaire). ABout this i suggest you to mention in your bibliography this work:
Galletti B, Sireci F, Mollica R, Iacona E, Freni F, Martines F, Scherdel EP, Bruno R, Longo P, Galletti. Vocal Tract Discomfort Scale (VTDS) and Voice Symptom Scale (VoiSS) in the Early Identification of Italian Teachers with Voice Disorders. Int Arch Otorhinolaryngol. 2020;24(3):e323-e329
It is essential to diagnose and correct comorbidities to obtain good results in bariatric surgery. Non-invasive tests are used for the subjective assessment of a patient's health and quality of life. Most often, these are questionnaires, the combined use of which is aimed at improving the detection of various disorders, e.g., voice disorders [12].
12. Galletti B, Sireci F, Mollica R, Iacona E, Freni F, Martines F, Scherdel EP, Bruno R, Longo P, Galletti. Vocal Tract Discomfort Scale (VTDS) and Voice Symptom Scale (VoiSS) in the Early Identification of Italian Teachers with Voice Disorders. Int Arch Otorhinolaryngol. 2020;24,3,e323-e329
Reviewer 2 Report
This study reports endoscopic data, referred to upper gastrointestinal tract, among patients candidate or not to bariatric surgery.
Since it is well-known that biopsy could be limited to detect Helicobacter pylori in relation to its gastric distribution, the authors should explain if in case of negativity they recommended a non-invasive test. If not, this should be reported as limitation of the study in the section discussion.
Since all patients eradicated Helicobacter pylori with a standard triple therapy, it seems that in this area there is a low resistance rate to antibiotics. The authors should report, if possible (if published), data on their area.
In table 1 some words are not translated in english "Choroby
współistniejące"
Please revise carefully english-language.
Line 253, it is not correct to report that triple therapy for Helicobacter pylori is the current standard. The actual approach depends on local antibiotic resistance. This should be clearly highlighted reporting that in the major part of European Countries the single caspule bismuth-based therapy is indicated and used (see the recent Nyssen et al. United Eur Gastroenterol J 2021;9:38-46).
Author Response
Responses for reviewer
Reviewer 2
- Since it is well-known that biopsy could be limited to detect Helicobacter pylori in relation to its gastric distribution, the authors should explain if in case of negativity they recommended a non-invasive test. If not, this should be reported as limitation of the study in the section discussion.
The limitation of the study was the lack of recommendation to perform a non-invasive test with a negative Helicobacter pylori biopsy result.
- Since all patients eradicated Helicobacter pylori with a standard triple therapy, it seems that in this area there is a low resistance rate to antibiotics. The authors should report, if possible (if published), data on their area.
The authors do not have data on antibiotic resistance after standard triple erradication therapy of Helicobacter pylori.
- In table 1 some words are not translated in english "Choroby
współistniejące"
Corrected: Comorbidities
- Please revise carefully english-language.
Yes, we use MDPI English editing Servis
- Line 253, it is not correct to report that triple therapy for Helicobacter pylori is the current standard. The actual approach depends on local antibiotic resistance. This should be clearly highlighted reporting that in the major part of European Countries the single caspule bismuth-based therapy is indicated and used (see the recent Nyssen et al. United Eur Gastroenterol J 2021;9:38-46).
All patients with a positive test for Helicobacter pylori underwent effective eradication therapy in accordance with the applicable standards and on the recommendation of the attending physician. However, in most European countries, therapy based on the use of a single capsule of bismuth is indicated and used [33].
- Nyssen, OP, Perez-Aisa A, Castro-Fernandez M, Pellicano R, Huguet JM, Rodrigo L, Ortuñ, J, Gomez-Rodriguez BJ, Pinto, RM, Areia M, Perona M, Nuñez O, Romano M, Gravina AG, Pozzati L, Fernandez-Bermejo M, Venerito M, Malfertheiner P, Fernanadez-Salazar L, Gasbarrini A, Vaira D, Puig I, Megraud F, O'Morain C, Gisbert JP. European Registry on Helicobacter pylori management: Single-capsule bismuth quadruple therapy is effective in real-world clinical practice. United European Gastroenterol J, 2021, 9, 38-46. https://doi.org/10.1177/2050640620972615
Round 2
Reviewer 2 Report
I have no further suggestions
Author Response
Thanks for your reviewing.